# Prediction of prognosis in immunoglobulin a nephropathy patients with focal crescent by machine learning

Xuefei Lin[1,2,3☯], Yongfang Liu[2,3☯], Yizhen Chen[1], Xiaodan Huang[1], Jundu Li[1], Yuansheng Hou[1], Miaoying Shen[1], Zaoqiang Lin[1,4], Ronglin Zhang[1,5], Haifeng Yang[6], Songlin Hong[7], Xusheng Liu[8]*, Chuan Zou[8,9]*

1 Second Clinical Medical College, Guangzhou University of Chinese Medicine, Guangzhou, Guangdong, China, 2 Department of Nephrology, Jiujiang Hospital of Traditional Chinese Medicine, Jiujiang, Jiangxi, China, 3 JiangXi Kidney Research Institute of Chinese Medicine, Jiujiang, Jiangxi, China, 4 Department of Nephrology, Shenzhen Hospital, Beijing University of Chinese Medicine, Shenzhen, China, 5 Department of Nephrology, Long Yan Hospital of Traditional Chinese Medicine, Longyan, Fujian, China, 6 Department of Pathology, Guangdong Provincial Hospital of Chinese Medicine, Guangzhou, Guangdong, China, 7 Fane Data Technology Corporation, Tianjin, China, 8 Department of Nephrology, Guangdong Provincial Hospital of Chinese Medicine, Guangzhou, Guangdong, China, 9 Guangdong-Hong Kong-Macau Joint Lab on Chinese Medicine and Immune Disease Research, Guangzhou University of Chinese Medicine, Guangzhou, Guangdong, China

☯ These authors contributed equally to this work.
* liuxu801@126.com (XL); doctorzc541888@126.com (CZ)

**Data Availability Statement:** All Python code and our study's data set files are available from the Figshare database (accession number(s) DOIs are

## Abstract

### Background and objectives

Immunoglobulin a nephropathy (IgAN) is the most common primary glomerular disease in the world, with different clinical manifestations, varying severity of pathological changes, common complications of crescent formation in different proportions, and great individual heterogeneous in clinical outcomes. Therefore, we aim to develop a machine learning (ML) based predictive model for predicting the prognosis of IgAN with focal crescent formation and without obvious chronic renal lesions (glomerulosclerosis <25%).

### Materials

We retrospectively reviewed biopsy-proven IgAN patients in our hospital and cooperative hospital from 2005 to 2017. The method of feature importance of random forest (RF) was applied to conduct feature exploration of feature variables to establish the characteristic variables that are closely related to the prognosis of focal crescent IgAN. Multiple ML algorithms were attempted to establish the prediction models. The area under the precision-recall curve (AUPRC) and the area under the receiver operating characteristic curve (AUROC) were applied to evaluate the predictive performance via three-fold cross validation (namely 2 training sets and 1 validation set).

### Results

RF was used to screen the important features, the top three of which were baseline estimated glomerular filtration rate (eGFR), serum creatine and triglyceride. Ten important

10.6084/m9.figshare.19127399 and 10.6084/m9.
figshare.19127342, respectively.).

**Funding:** The study was supported by the
Research Project for Practice Development of
National TCM Clinical Research Bases (Project No.
JDZX2015202), the 2020 Guangdong Provincial
Science and Technology Innovation Strategy
Special FundGuangdong-Hong Kong-Macau Joint
Lab(2020B1212030006), Industry Special of the
State Administration of traditional Chinese
Medicine(201407005) and Guangzhou University
of Traditional Chinese Medicine double first-class
and high-level university discipline collaborative
innovation team project (2021xk66).

features were selected as important predictors for modeling on the basis of data-driven and medical selection, predictors include: age, baseline eGFR, serum creatine, serum triglycerides, complement 3(C3), proteinuria, mean arterial pressure (MAP) and Hematuria, crescents proportion of glomeruli, Global crescent proportion of glomeruli. In a variety of ML algorithms, the support vector machine (SVM) algorithm displayed better predictive performance, with Precision of 0.77, Recall of 0.77, F1-score of 0.73, accuracy of 0.77, AUROC of 79.57%, and AUPRC of 76.5%.

## Conclusions

The SVM model is potentially useful for predicting the prognosis of IgAN patients with focal crescent shape and without obvious chronic renal lesions.

## Introduction

IgA nephropathy (IgAN) is the most common primary glomerular disease in the world, accounting for more than 40% of primary glomerular diseases. Its adult incidence rate is more than 2.5/ 100,000 / year, which is the main cause of end stage renal disease (ESRD) [1–3]. 20–40% of patients with IgAN will develop ESRD within 10–20 years [4]. In the latest MEST-C Oxford typing in 2017 [5], crescent is an independent predictor of prognosis in patients with IgAN, and there is also a proportional dependence between crescent proportion and prognosis. Therefore, the prognosis of IgAN with different proportion of partial crescent formation is different, and the proportion of crescent can be included in the prognosis study. However, the variable of crescent is not included in the two prediction models recommended by the 2021 KDIGO guidelines, due to the crescent is highly related to race/ethnicity and the use of immunosuppressants after biopsy [6]. Of note, this problem could be solved by the importance ranking function of RFs, from which the importance score of each factor would be applied to reflect its own contribution. According to importance scores, the predictive value of clinicopathological parameters should be explored to predict ESRD for a more suitable prective model for Chinese IgAN patients. This indicates that there is a clear need for a predictive model that includes crescents as an important predictor to predict disease progression in IgAN.

Many previous prediction models for IgAN have used standard modeling with a small number of predefined the risk of demographic, clinical, and pathological variables predicting the progression of IgAN to end-stage renal disease [6–10]. However, previous studies have mostly used standard statistical methods, such as univariate and multivariate Cox regression models and proportional hazard models, which only evaluate the relationship between variable quantum sets and ESRD progress, and potentially ignore the important interactions between variables and their effects on ESRD progress. Compared with conventional statistical methods, machine learning (ML) has better ability to identify variables related to clinical outcomes, better predictive performance, better complex relationship modeling ability, robustness to data noise, and the ability to learn from multiple data modules. ML's application in furthering nephrology research and practice are myriad [11, 12]. Recently, Random forest and ANN model have been applied to predict progression to ESRD in IgAN patients [13, 14]. ML algorithms display better predictive performance and lower errors.

In this study, multiple ML algorithms were applied to predict ESRD progression in IgAN patients. The purpose of this study is to successfully identify patients at high risk of progression to ESRD to facilitate early and effective treatment. Since the updated Oxford classification

included crescent, the prediction model has included the crescent into the prediction model of predictive variables. However, they only simply include C0, C1 and C2, without subdividing the size, proportion and nature of the crescent. Therefore, this study innovatively incorporated the crescent index of different size, proportion and nature into the prognosis study of IgAN.

## Materials and methods

### Study population

In this study, the 662 biopsy-proven IgAN patients were collected from Guangdong Provincial Hospital of Chinese Medicine and Shanxi Traditional Chinese Medicine Hospital between May 2005 and November 2017. The inclusion criteria were as follows: (1) Age > 18 years old;(2) Patients with biopsy-proven primary IgAN;(3) glomerulosclerosis proportion < 25%;(4) Patients were followed for more than 12 months unless ESRD occurred within 12 months. Patients who met any of the following criteria were excluded: (1) insufficient clinical and pathological data;(2) Patients with secondary causes of mesangial IgA deposits, such as IgA vasculitis and systemic lupus erythematosus, or those with comorbid conditions, such as diabetes mellitus, were excluded;(3) atypical IgAN, such as crescentic IgAN;(4) tubulointerstitial fibrosis caused by drugs and ischemia;(5) a biopsy specimen with less than 8 total glomeruli. This study was approved by the research ethics committee of Guangdong Provincial Hospital of Chinese Medicine, IRB number: B2016-155-01. This study was a retrospective study, all data were completely anonymized, and the ethics committee waived the requirement for informed consent.

### Dataset collection and definitions of variables

In this study, baseline demographics, clinical and pathology data were collected for all patients during renal biopsies, including age, gender, mean arterial pressure (MAP) defined as diastolic pressure plus one-third of the pulse pressure, 24-hour protein excretion and estimated glomerular filtration rate (eGFR) calculated by the Chronic Kidney Disease Epidemiology (CKD-EPI) Collaboration equation. Regardless of the duration and dose, the type of immunosuppression or renin-angiotensin-aldosterone system (RAAS) blockades therapy that the patient received was recorded. Immunosuppression was defined as treatment with corticosteroids and/or corticosteroid-sparing agents (including cyclophosphamide, azathioprine, mycophenolate, cyclosporine or tacrolimus). RAAS blockades included any exposure to angiotensin-converting enzyme inhibitor and/or angiotensin receptor blocker after biopsy. The updated Oxford Classification (MEST-C) for IgAN was applied in this study [5]. Renal biopsy samples from all patients were examined by pathologist and nephrologist. The crescent is subdivided according to the volume, composition and proportion of the crescent. The volume of the crescent body is defined as the large crescent body accounting for 50% or more of the renal sac volume and the small crescent body accounting for 50% or less of the renal sac volume. The components of crescent body can be divided into cellular crescent, cellular fibrous crescent and fibrous crescent. The cellular crescent consists of > 75% cells and < 25% fibrous matrix. The fibrous cellular crescent consists of 25%-75% of the cells and the remaining fibrous matrix. Extracapillary fibrosis of fibrous crescents consists of > 75% matrix and < 25% cells. The crescent ratio is defined as the proportion of the number of glomeruli with crescents in the total number of glomeruli, and the cell / fibrous cell / fibrous crescent is evaluated according to the relative ratio. ESRD was defined as eGFR<15 mL/min/1.73 m2 for more than 3 months or initiation of dialysis or transplantation. In this study, we defined clinical outcome: the combined event (Doubling of serum creatinine, 50% reduction in eGFR, 15% reduction in eGFR within 1 year, 30% reduction in eGFR within 2 year, ESRD or death) after diagnostic kidney biopsy.

## ML algorithms

In the study, a variety of representative supervised classification ML algorithms were applied to build models. Three prediction models, Support Vector Machine (SVM) model, Random Forest (RF) Model and Naïve Bayes (NB) Model were used to build a prediction model based on the variables selected above. SVM, RF and NB are "black box"models, where the function connecting the predictor variables with response is unclear to the user. The receiver operating characteristic (ROC) curve, precision-recall curve (PRC) and lift curve were used to assess the predictive performance as previously described.

## Feature selection and model construction

In this study, 39 clinical, pathological and demographic parameters were applied to predict the progression status of IgAN. To explore the better models, random forest algorithm which can assess the importance of all variables was adopted to evaluate the importance rankings of correlated predictive factors related to the prognosis of IgAN. In order to compute the importance of each predictive feature, all the features were used in the RF method. It is not enough to use the rank of important features of random forest for feature selection, but also to consider the characteristics of clinical specialty. Thus, in this study, we established and further evaluated the performance of 2 kind of models without and with crescent in a cohort of IgAN patients from China. Additionally, ML models (random forest classifier, support vector machine, Naïve Bayes, etc.) were cross-verified with 3 fold cross-validation (namely 2 training sets and 1 validation set). The detailed process of model cross-validation is shown in Fig 1. The ML algorithms were implemented using Python 3.8.5 (https://www.python.org) with scikit-learn (https://scikit-learn.org/stable/).

## Statistical analysis

Continuous variables were presented as the means and standard deviations for normal distributions and as medians and interquartile ranges for non-normal distributions. The categorical

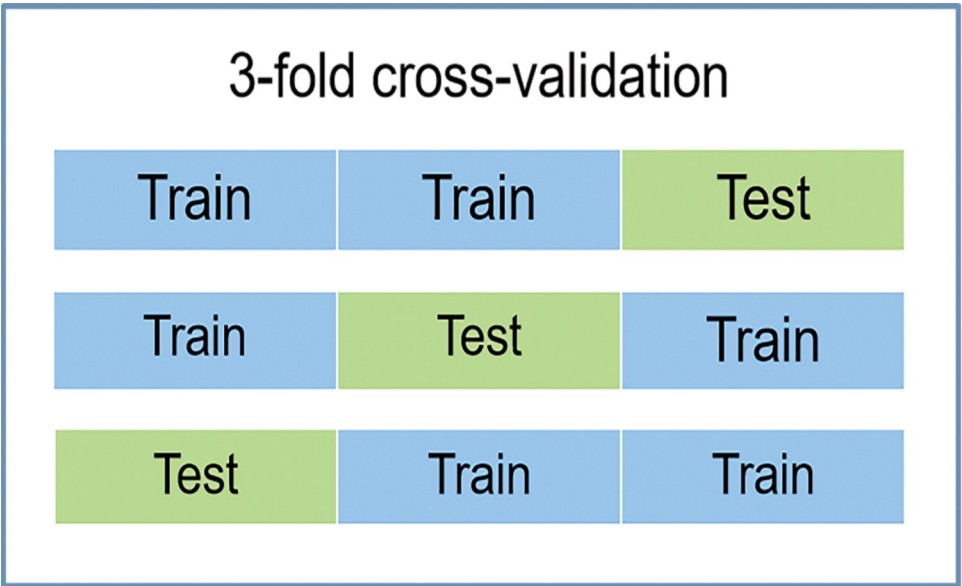

**Fig 1. All ML models were cross-verified with 3 fold cross-validation.**

variables were presented as the number and percentages. The independent sample t test was used for normally distributed continuous variables, Mann-Whitney U test for Non-normally distributed continuous variables and Pearson Chi-Square test or Fisher' s exact probability test for categorical variables. The P value<0.05 was considered statistically significant. Statistical analyses were performed using IBM SPSS Statistics software (Version 25.0. IBM Corporation, NY, USA).

## Results

### Clinical and pathological characteristics of the population

From May 2005 and November 2017, 374 biopsy-proven IgAN patients were recruited eventually (Fig 2), whose characteristics are shown in Table 1. In our cohort, 17.6% of the 374 IgAN patients reached the combined event within a median follow-up time of 32.99 (25.86–54.68) months.

The demographic, clinical, pathologic and treatment characteristics of patients at the time of biopsy with and without progression to the combined event were retrospectively compared. The median age of the enrolled patients in this cohort at IgAN diagnosis was 31(26–40) years, of whom 175(46.8%) were male. At the time of renal biopsy, patients had the median urinary protein excretion of 0.85(0.43–1.57) g/24 h and the mean eGFR was 108.3±39.49 mL/min/1.73 m$^2$. The median MAP was 93.17(84.25–100) mmHg, and 31% (116) of the patients presented with hypertension history. In total, 119 (31.8%) patients had crescents in glomeruli. Of these patients, 105 (28.1%) had crescents in less than 1/4 of glomeruli (C1 group), and 14 (3.7%) had crescents in more than 1/4 of glomeruli (C2 group). Regarding MEST Oxford scores in all patients, 82.4% were M1, 13.4% were E1, 53.2% were S1, and 14.5% were T1/T2. After diagnosis, 247 (66%) patients received RAAS blockade, which include Angiotensin-Converting Enzyme Inhibitors (ACEI) and Angiotensin Receptor Blockers (ARB). During the course, 140 (37.4%) patients received immunosuppressants including corticosteroids, cyclophosphamide, ciclosporin, mycophenolate mofetil and tripterygium glycosides, as appropriate. There was no significant difference in the proportion of patients who were treated with RAAS blockade

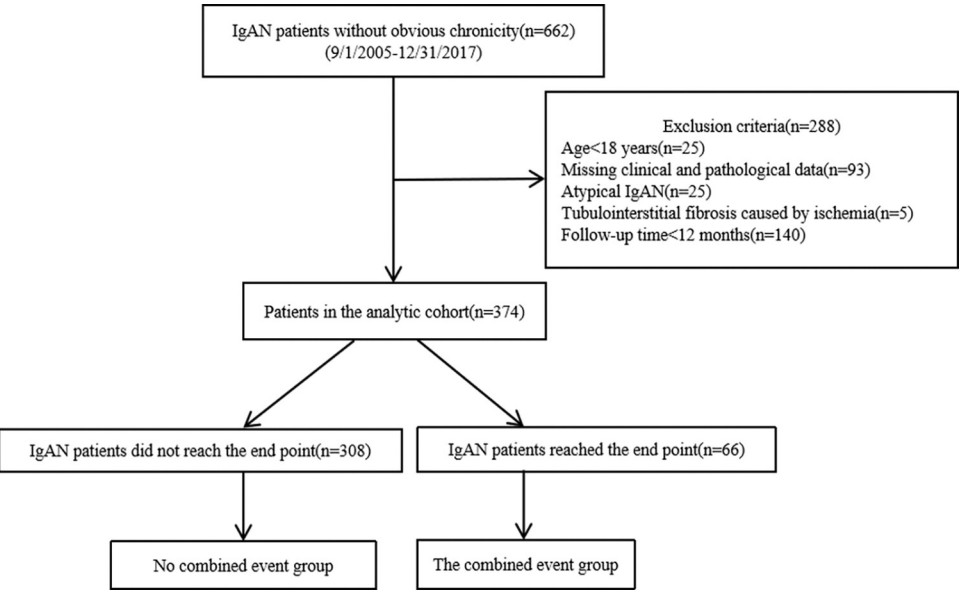

**Fig 2. Enrollment of IgAN patients in our cohort.**

**Table 1. Baseline cohort characteristics.**

| Factors | Overall (N = 374) | None-Endpoint (N = 308) | Endpoint (N = 66) | P value |
|---|---|---|---|---|
| Male, n (%) | 175(46.8) | 153(49.68) | 22(33.33) | 0.016* |
| Age(years) | 31(26–40) | 31(26–38.75) | 31.5(25–46) | 0.502 |
| Follow up, n (%) | 32.99(25.86–54.68) | 33.13(26.04–55.71) | 31.62(24.73–50.19) | 0.288 |
| Disease course, months | 7(1–24) | 7.5(1–24) | 5.5(1–24) | 0.821 |
| eGFR, mL/min/1.73m$^2$ | 108.3±39.49 | 106.92±35.76 | 114.75±53.46 | 0.259 |
| Serum creatine, umol/L | 80.7±29.74 | 81.14±28.47 | 78.66±35.28 | 0.54 |
| Proteinuria, g/24h | 0.85(0.43–1.57) | 0.8(0.43–1.49) | 1.03(0.48–2.32) | 0.055 |
| Hematuria (red blood cells/high-power field) | 51(22.75–146.4) | 51(23–145.5) | 51(19.95–173.75) | 0.998 |
| BUN, mmol/L | 4.77(4–5.78) | 4.8(4.04–5.77) | 4.57(3.69–5.8) | 0.455 |
| Uric Acid, mmol/L | 341.5(280–414.25) | 343.5(280–416) | 335(278–406) | 0.752 |
| Cholesterol, mmol/L | 4.6(4–5.39) | 4.6(4–5.38) | 4.69(4.11–5.46) | 0.562 |
| Triglyceride, mmol/L | 1.2(0.9–1.76) | 1.2(0.89–1.7) | 1.33(0.88–2.31) | 0.224 |
| HDL-C, mmol/L | 1.27(1.05–1.53) | 1.24(1.03–1.55) | 1.35(1.09–1.53) | 0.338 |
| LDL-C, mmol/L | 2.82(2.32–3.5) | 2.89(2.34–3.48) | 2.76(2.22–3.51) | 0.752 |
| Blood glucose | 4.85(4.42–5.1) | 4.85(4.4–5.1) | 4.79(4.44–5.11) | 0.731 |
| TP, g/L | 67(62–71.53) | 67.3(62.98–71.98) | 65.5(59.38–70.35) | 0.031* |
| Serum albumin, g/L | 40.9(37.18–43.9) | 41.1(37.7–44.15) | 38.8(34.28–42.7) | 0.005* |
| Serum IgA, g/L | 3.05(2.46–3.5) | 3.05(2.45–3.54) | 3.05(2.47–3.49) | 0.959 |
| Serum C3, g/L | 1.02(0.9–1.11) | 1.02(0.9–1.1) | 1.02(0.93–1.14) | 0.585 |
| SBP, mmHg | 120(110–130) | 120(110–130) | 121.5(112.75–134) | 0.149 |
| DBP, mmHg | 79.5(70–86) | 80(70–85) | 78.5(70–89.25) | 0.419 |
| MAP, mmHg | 93.17(84.25–100) | 92.84(83.67–100) | 93.67(86.5–102.84) | 0.239 |
| Hypertension (%) | 116(31) | 93(30.19) | 23(34.85) | 0.458 |
| Diabetes (%) | 6(1.6) | 4(1.3) | 2(3.03) | 0.287 |
| Hepatitis (%) | 28(7.5) | 26(8.44) | 2(3.03) | 0.195 |
| CVD (%) | 1(0.3) | 1(0.32) | 0(0) | 1 |
| Smoke (%) | 15(4) | 12(3.9) | 3(4.55) | 0.735 |
| Alcohol (%) | 10(2.7) | 9(2.92) | 1(1.52) | 1 |
| M1 (%) | 308(82.4) | 249(80.84) | 59(89.39) | 0.098 |
| E1 (%) | 50(13.4) | 39(12.66) | 11(16.67) | 0.386 |
| S1 (%) | 199(53.2) | 157(50.97) | 42(63.63) | 0.061 |
| T1 (%) | 46(12.3) | 35(11.36) | 11(16.67) | 0.209 |
| T2 (%) | 5(1.3) | 3(0.97) | 2(3.03) | 0.197 |
| C1 (%) | 105(28.1) | 85(27.6) | 20(30.3) | 0.556 |
| C2 (%) | 14(3.7) | 10(3.25) | 4(6.06) | 0.42 |
| RAAS blockade (%) | 247(66) | 206(66.9) | 41(62.1) | 0.458 |
| Immunosuppressant (%) | 140(37.4) | 108(35.1) | 32(48.5) | 0.041* |

The demographic, clinical, laboratory data and treatment of the IgAN patients. C3, complement 3; TP, total protein; MAP, mean arterial pressure; eGFR, estimated glomerular filtration rate; LDL-C, low density lipoprotein cholesterol; HDL-C, high density lipoprotein cholesterol; BUN, blood urea nitrogen; SBP, Systolic blood pressure; DBP, Diastolic blood pressure; CVD, cardiovascular disease, RAAS, renin-angiotensin-aldosterone system. Immunosuppressants include Steroids, cyclophosphamide, ciclosporin, mycophenolate mofetil and others.

* P < 0.05

between the none-endpoint group and the endpoint group (66.9% vs 62.1%, p = 0.458). The proportion of Endpoint group who were treated with immunosuppressants was more higher than the none-endpoint group (48.5% vs 35.1%, p = 0.041). Other demographic, clinical, and laboratory data of the IgAN patients are shown in Table 1.

## Feature importance and selection

To identify crucial predictors of the combined event, we employed the Random Forest (RF) method to calculate the feature scores of all features (S1 Table shows all features). The feature selection method of our modeling considers the following principles to select the features that participate in the modeling: (1) The top features found by the feature selection algorithm. (2) The selected features cover different aspects as far as possible, such as patient pathological characteristics, clinical characteristics, epidemiological characteristics, and so on. (3) The selected features are as independent as possible, that is, to minimize the strong correlation between multiple variables. (4) Generally speaking, the amount of data of a mathematical model should be at least 10 times the number of independent variables of the model, on the other hand, the number of features involved in modeling should not be too much. The mathematical model established by too many variables can be poorly explained. (5) Focus on the principles and practical experience of the medical profession. Therefore, based on the above principles, we selected the top ten features including: baseline estimated GFR, serum creatine, serum triglycerides, proteinuria, MAP and Hematuria, C3, age, crescents proportion of glomeruli, Global crescent proportion of glomeruli. All of these features displayed a strong correlation with the combined event (Fig 3 shows the feature importance). A total of 10 prioritized features were selected as important predictors for modeling on the basis of the ranking of important features and medical selection, as shown in Table 2. The establishment of prediction models of predictors can be roughly divided into three aspects: patient epidemiological characteristics: age; clinical features: baseline estimated GFR, serum creatine, serum triglycerides, C3, proteinuria, MAP and Hematuria (red blood cells/high-power field); pathological findings: crescents proportion of glomeruli, Global crescent proportion of glomeruli.

## ML models establishment and evaluation

In the study, above ten important features were applied to IgAN with crescent models, however, the first eight important features were applied to IgAN without crescent models. For the selection of a better predictive model, several widely applied ML algorithms were compared, including support vector machine (SVM), Random Forest (RF), and Naïve Bayes (NB), by using the receiver operating characteristic curve and precision-recall curve. In IgAN with crescent models, the AUROCs of the SVM model, RF model and NB model are 0.7957, 0.6443 and 0.7078, respectively (Fig 4). The AUPRCs of the SVM model, RF model and NB model are 0.765, 0.472 and 0.637, respectively (Fig 5). In IgAN without crescent models, the AUROCs of the SVM model, RF model and NB model are 0.831, 0.7041 and 0.5959, respectively (Fig 6). The AUPRCs of the SVM model, RF model and NB model are 0.716, 0.567 and 0.567, respectively (Fig 7). Receiver operating characteristic curves and precision-recall curves both show the superiority of the SVM model.

Table 3 summarizes the IgAN with crescent model metrics, including precision, recall, F1-score (higher is better), AUROC, and AUPRC (higher is better). The Support Vector Machine model exhibited the highest Precision (0.77), Recall (0.77), F1-score (0.73), Accuracy (0.77), AUROC (0.7957) and AUPRC (0.765). Table 4 summarizes the IgAN without crescent model metrics. The Support Vector Machine model exhibited the highest Precision (0.78), Recall (0.68), F1-score (0.56), Accuracy (0.68), AUROC (0.831) and AUPRC (0.716).

## Performance evaluation of excellent models with Lift

The Lift curve is one of the most commonly used methods for ML classification. Lift reflects how many times the accuracy of prediction is improved compared with random selection without prediction model. Lift reveals the effect of the prediction model, it should be as steep as possible. To obtain a more reliable evaluation of the performance of the prediction model,

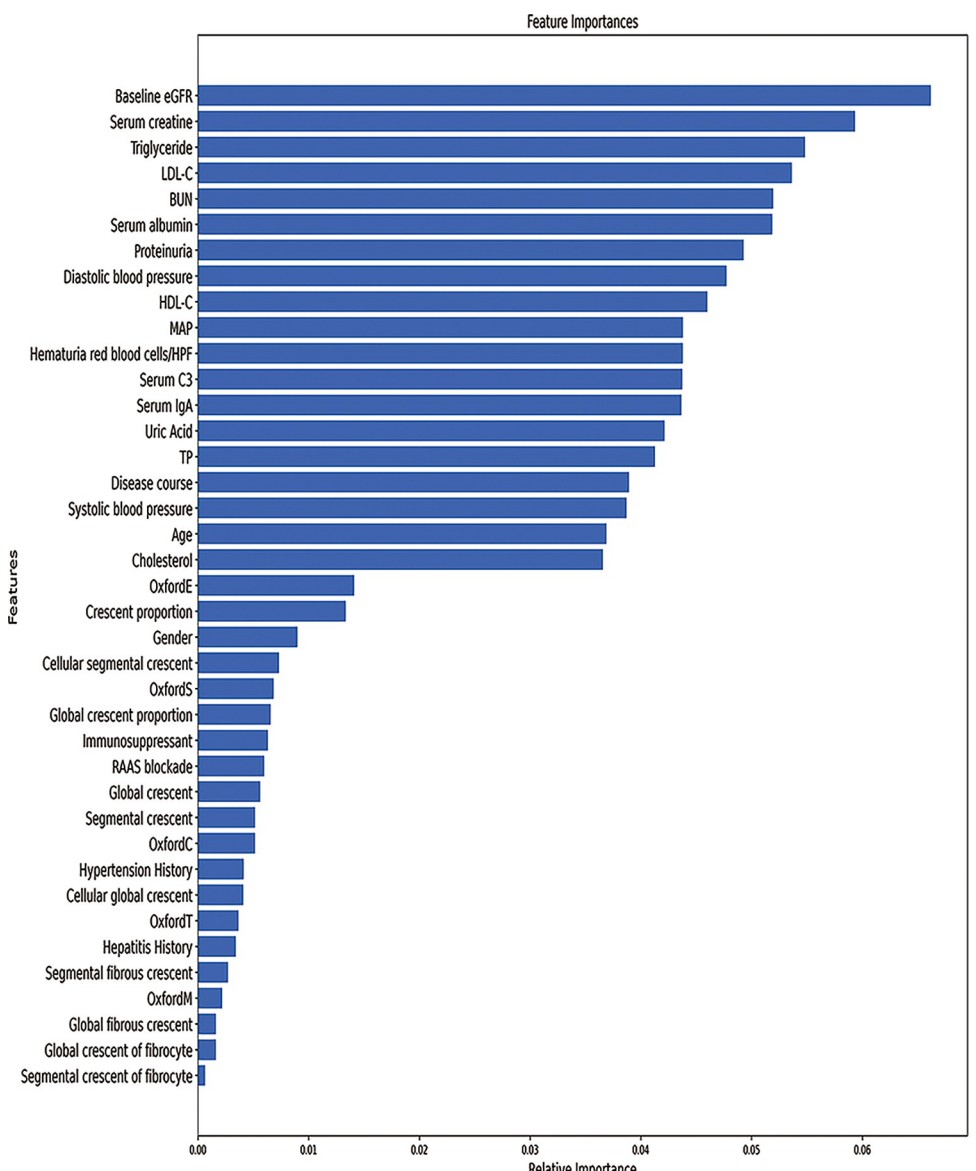

**Fig 3. Contribution of the included features of the combined event in IgAN patients.** HDL-C, High density lipoprotein cholesterol, LDL-C, Low density Lipoprotein cholesterol, TP, Total serum protein.

this study will comprehensively apply ROC curves and Lift curves to verify the performance of the model based on different algorithms. In IgAN with crescent models, the larger the Lift value of the SVM prediction model is, the better the model effect. As shown in Fig 8, the SVM model predicted the conbined endpoint event of IgAN, Lift = 3.65, which is 3.65 times more accurate than random prediction, and none conbined endpoint event of IgAN, Lift = 1.38, which is 1.38 times more accurate than random prediction. The Lift curve basically shows a downward trend, also suggesting that the SVM model has good prediction performance.

## Models calibration

The Calibration of the prediction model is an important index to evaluate the accuracy of a disease risk model in predicting the probability of an individual outcome event in the future. It

**Table 2. Predictors selected using random forest and the corresponding feature importance score.**

| Features | Importance score |
|---|---|
| Baseline eGFR, ml/min per 1.73m$^2$ | 0.066177 |
| Serum creatine, mmol/L | 0.059347 |
| Serum triglycerides, mmol/L | 0.054830 |
| Proteinuria, g/d | 0.049275 |
| MAP, mm Hg | 0.043798 |
| Hematuria (red blood cells/high-power field) | 0.043790 |
| Serum C3, g/L | 0.043743 |
| Age at biopsy, years | 0.036900 |
| Crescent proportion of glomeruli, % | 0.013346 |
| Global crescent proportion of glomeruli, % | 0.006574 |

eGFR, estimated glomerular filtration rate; MAP, mean arterial pressure; C3, complement 3.

reflects the consistency between the predicted risk and the actual occurrence risk of the model, so it can also be called consistency. Good calibration indicates that the prediction model has high accuracy; poor calibration indicates that the model may overestimate or underestimate the risk of disease. As shown in Figs 9 and 10, relatively speaking, the blue Random Forest model and the black Support Vector Machine Model calibration curve are better.

## Discussion

Crescents have been implicated as an important marker of poor prognosis of IgAN. However, a validation study of crescent failed to validate the increased risk of renal function progression

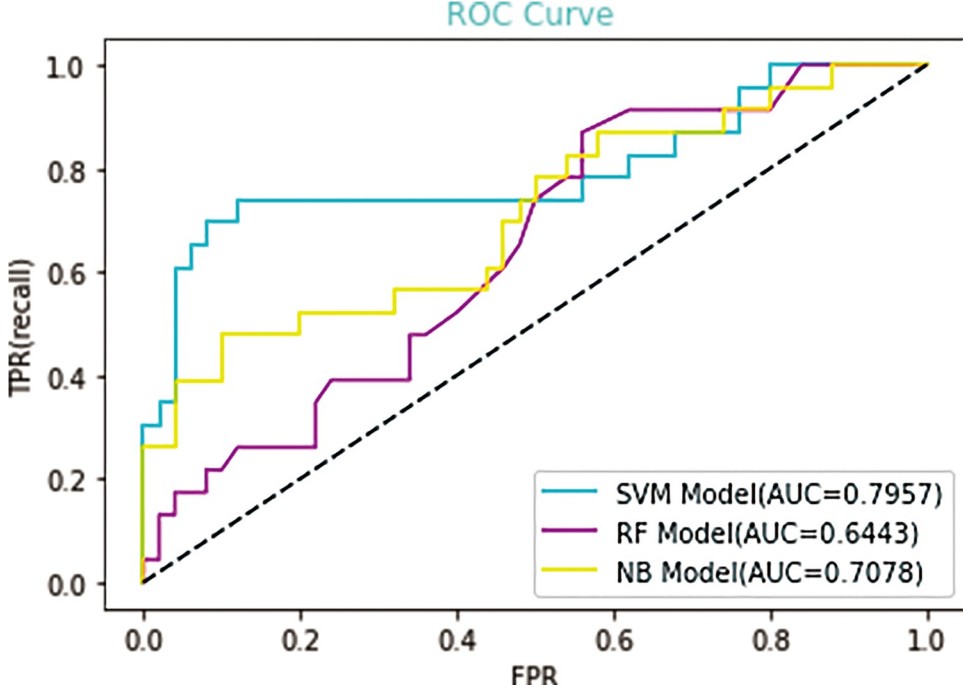

**Fig 4. Receiver operating characteristic (ROC) curves of the three candidate models for the prognosis of IgAN.** AUC, area under the curve.

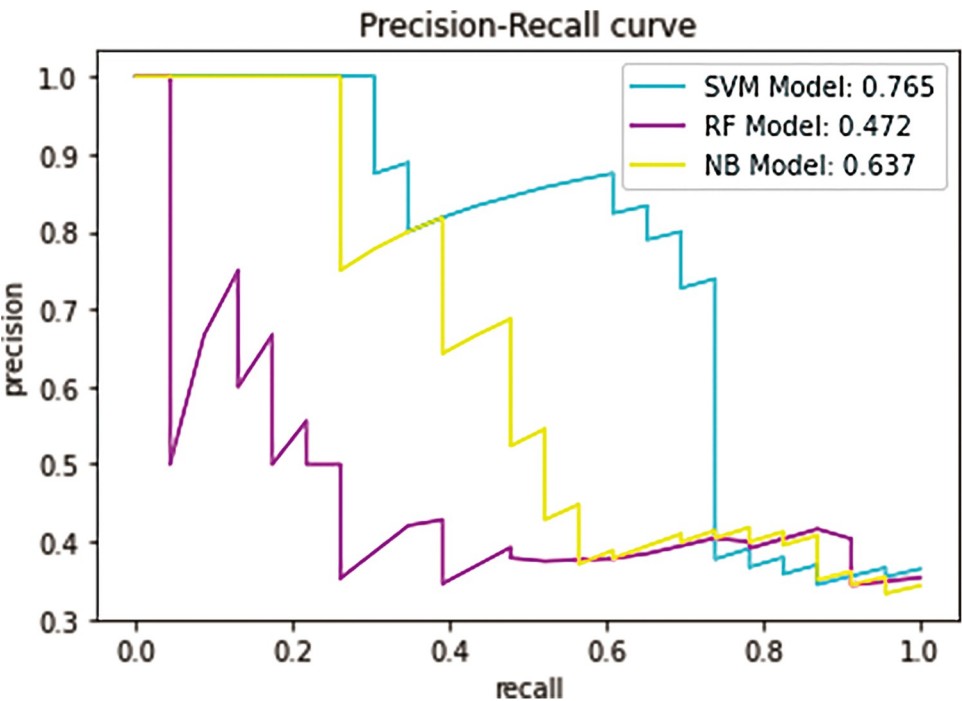

**Fig 5. Precision-recall curves of the three candidate models for the prognosis of IgAN.**

in Chinese IgAN patients in the C1 or C2 groups compared to the C0 group [15]. The discrepant findings may be due to the different definitions of outcomes. In addition, considering the inherent nature of crescents, some IgAN patients with more crescents may occur in an early or

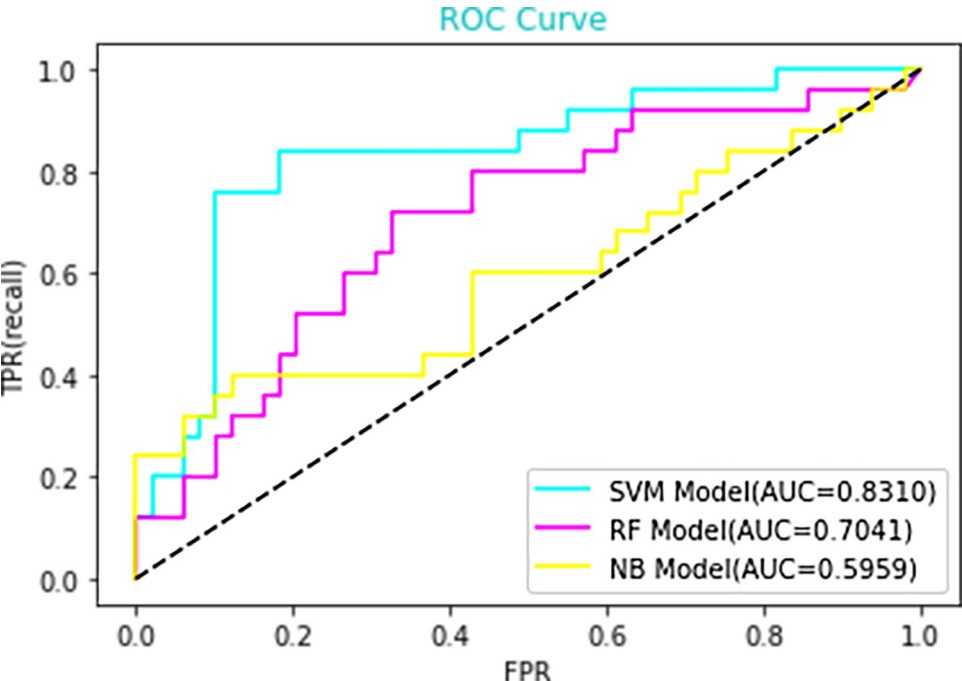

**Fig 6. Receiver operating characteristic (ROC) curves of the three candidate models for the prognosis of IgAN without 'Crescent proportion' and 'Global crescent proportion'.**

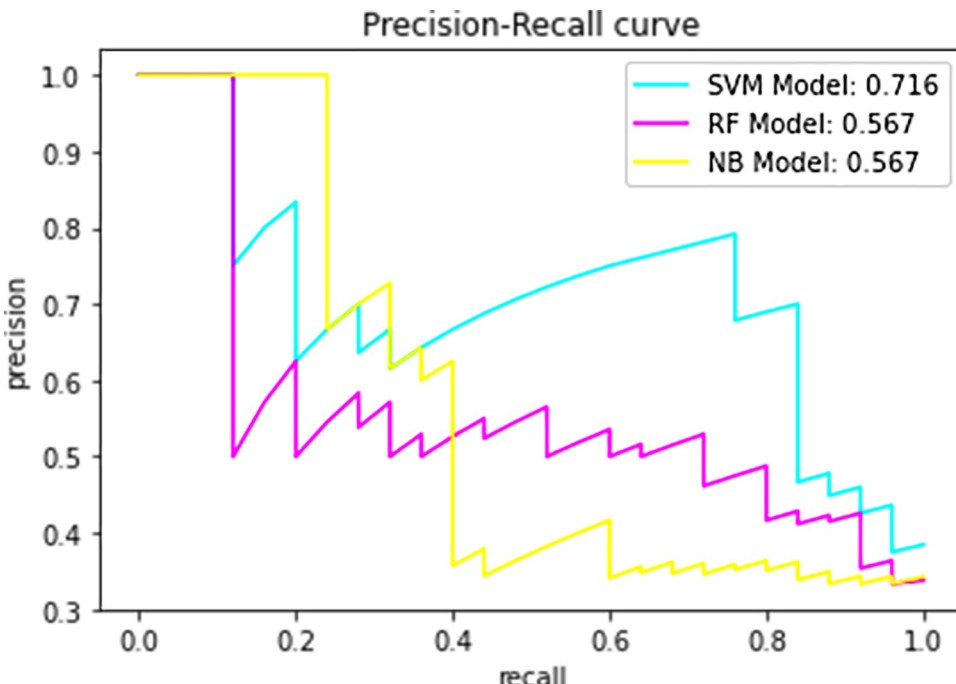

**Fig 7. Precision-recall curves of the three candidate models for the prognosis of IgAN without 'Crescent proportion' and 'Global crescent proportion'.**

acute stage of renal damage. Therefore, we defined clinical outcome: the combined event after diagnostic kidney biopsy. However, it has been reported that the proportion of glomerulo-sclerosis >25% as chronic pathological lesions which may interfere with crescents as active lesions on the prognosis of IgAN, which are both associated with a decreased renal survival rate in IgAN patients [16]. Previous studies have shown that the pathological types of renal biopsy in IgAN patients are mainly non-obvious chronicity lesions (glomerulosclerosis <25%, T score<2) [6, 16, 17]. However, in IgAN patients without obvious chronic lesions, few predic-tion model have studied the effect of crescent index of different size, proportion and nature into the prognosis study of IgAN. Therefore, 374 patients with IgAN without obvious chronic

**Table 3. Summary of the comparison of IgAN with 'Crescent proportion' and 'Global crescent proportion' model performance.**

| Prediction model | Precision | Recall | F1-score | Accuracy | AUROC | AUPRC |
|---|---|---|---|---|---|---|
| Support Vector Machine | 0.77 | 0.77 | 0.73 | 0.77 | 0.7957 | 0.765 |
| Random Forest | 0.69 | 0.70 | 0.61 | 0.70 | 0.6443 | 0.472 |
| Naïve Bayes | 0.74 | 0.74 | 0.69 | 0.74 | 0.7078 | 0.637 |

**Table 4. Summary of the comparison of IgAN without 'Crescent proportion' and 'Global crescent proportion' model performance.**

| Prediction model | Precision | Recall | F1-score | Accuracy | AUROC | AUPRC |
|---|---|---|---|---|---|---|
| Support Vector Machine | 0.78 | 0.68 | 0.56 | 0.68 | 0.831 | 0.716 |
| Random Forest | 0.65 | 0.68 | 0.63 | 0.68 | 0.7041 | 0.567 |
| Naïve Bayes | 0.70 | 0.72 | 0.70 | 0.72 | 0.5959 | 0.567 |

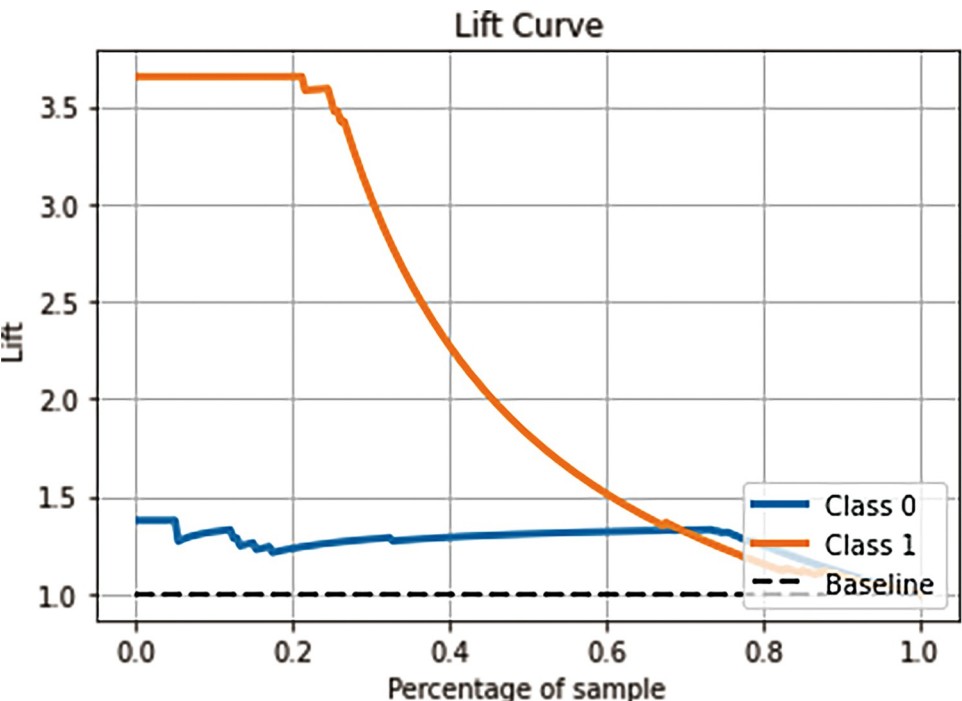

**Fig 8. The Lift curve with Support Vector Machine model.** "Class 0" indicates IgAN patients with none conbined endpoint progression, and "Class 1" indicates IgAN patients with the conbined endpoint progression.

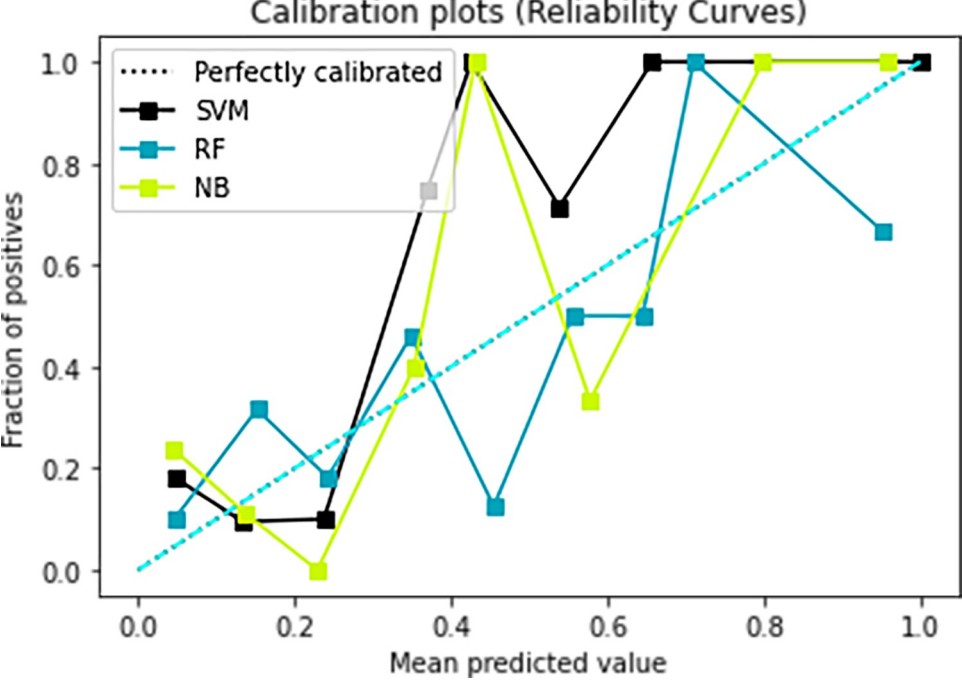

**Fig 9. Calibration plots of the three candidate models for the prognosis of IgAN with 'Crescent proportion' and 'Global crescent proportion'.**

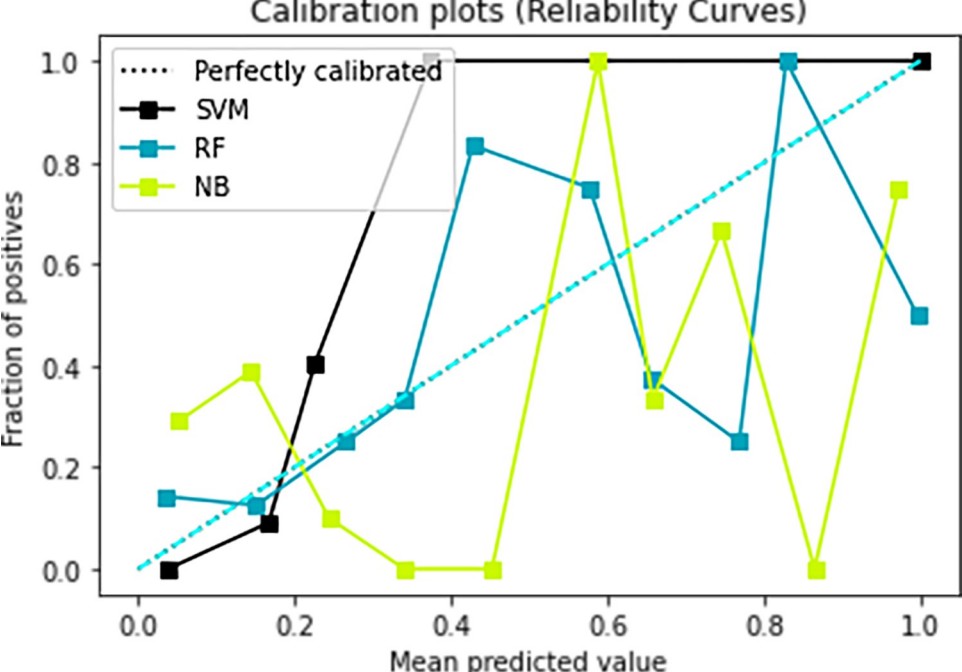

**Fig 10. Calibration plots of the three candidate models for the prognosis of IgAN without 'Crescent proportion' and 'Global crescent proportion'.**

lesions were retrospectively analyzed and multiple ML algorithms were applied to explore a useful and practical predictive model.

Some baseline characteristics of patients at renal biopsy between negative and positive end-point groups are significantly different so it is possible to use baseline characteristics at renal biopsy to predict the conbined event of IgAN patients. To date, several clinical and pathological parameters have been associated with a high risk of kidney disease progression in IgAN. Previously identified risk parameters for IgAN include gender [13, 18, 19], age [6, 13, 18–20], baseline serum creatinine concentration [13, 18–20], eGFR [6, 18], SBP [6, 13, 18–20], DBP [6, 13, 18–20], proteinuria [6, 13, 18–20], hematuria [20], serum UA concentration [20], serum albumin concentration [18], treatment type[6, 13, 18] and histology grading [6, 13, 18–20].

In this study, as the risk factors identified above, the clinical and pathological factors included baseline eGFR, serum creatine, Serum triglyceride, proteinuria, MAP, hematuria, age at biopsy, Serum C3, Crescent proportion of glomeruli and Global crescent proportion of glomeruli. Based on the random forest algorithm, the top two important features were baseline eGFR and serum creatine, which are the demonstrated strong predictors for IgAN prognosis. Hypertriglyceridaemia at the time of diagnosis, which may have a role in tubulointerstitial lesions, is the important independent risk factor of poor outcome in IgAN [21]. In addition to the well known risk factors, proteinuria, age and MAP, hematuria was independently associated with IgAN progression [22]. It is reported that the formation of crescents is related to the degree of mesangial C3 deposition. Low serum levels of complement C3 is often associated with poor renal outcomes in IgAN [23]. Consistent with our previous study, global crescent and Serum C3 are the independent risk factor for IgAN progression and poor renal outcome in IgAN patients without significant chronic kidney damage [24]. Further more, Latest study show that crescent is independently associated with higher mortality in IgAN [25]. Thus, all of

these ten prioritized predictive features described above have clinically reasonable explanations.

In our study, several prevailing algorithms were applied to identify severe progress or poor prognosis of IgAN. Our raw dataset showed an obvious imbalance with 66 target events (the combined event) and 308 negative samples. Such imbalance is very common in clinical research. Therefore, we adopted the method of three-fold cross-validation and over-sample to improve the stability and generalization of the model and reduce the effect of imbalance. Comparing 3 ML models, SVM model outperformed the other 2 ML algorithms including Random Forest, and Naïve Bayes. To evaluate the prediction and accuracy of various ML models, we calculated and compared areas under the receiver operating characteristic curve (AUROC). Although SVM of IgAN without crescent model has higher AUROC, SVM of IgAN with crescent model has higher AUPRC and F1 values. AUROC is a general metric for model selection, however it is not the only reference. In clinical practice, particularly for our imbalanced dataset, the AUPRC and F1-score are more practical evaluation indicators for ML. Based on this requirement, we preferred the SVM of IgAN with crescent model rather than other models. SVM is an algorithm for identifying a high-dimensional boundary that distinctly classifies data points.

Notable strengths of our study include our choice to select patients who were without obvious chronic renal lesions (glomerulosclerosis <25%) at the time of biopsy which can clearly identify the effect of crescents on the prognosis of IgAN. In our study, two-center IgAN cohort, we excluded crescentic IgAN in order to predict the progress of IgAN with a focal crescent shape. Besides, we identified the important features using a more objective ML approach. Further more, in addition to the definitive outcome of ESRD or death, we also incorporated a 50% reduction in eGFR, doubling of serum creatinine, 15% reduction in eGFR within 1 year, 30% reduction in eGFR within 2 year, in our combined event to implement variable selection, which is more appropriate to evaluate the prognosis of patients with IgAN with manifestations of different severity. Lastly, and perhaps most importantly, we apply ML algorithms, which can build complex models and make accurate decisions rather than traditional statistical methods. The strength of our study is that we selected important predictors for modeling on the basis of feature scores and medical selection to avoid ignoring non-statistically-significant parameters or non-clinical parameters.

Our study was subject to some limitations. First of all, only Chinese patients were included in the model, and prediction for other populations was not evaluated. Secondly, our cohort was not large enough and our data was lost and unbalanced. As such, predictive ability was impaired by the relatively small numbers of positive events resulting from data imbalance. Thirdly, due to the limitation of the retrospective study design, the duration and dosage of IS therapy were not collected, more prospective studies with a larger cohort are needed to support the present findings, retrospective analyses alone are not sufficient to determine the treatment choice for patients with IgAN. Further more, external validation is required to prevent overfitting. Finally, this model has been developed and tested in retrospective cohorts which could not show the effect of the model in guiding treatment. However, the current prediction model is an effective and simple method to predict the progression of IgAN patients to the conbined event.

## Conclusion

In conclusion, ML algorithms exhibit an excellent predictive performance for IgAN patients with a focal crescent shape. Among these algorithms, support vector machine model, show the higher sensitivity, AUROC and Lift, can be used to predict the prognosis of IgAN patients with a focal crescent shape. However, we also identified that eGFR, serum creatine, Serum triglyceride, proteinuria, MAP, hematuria, age, Serum C3, Crescent proportion of glomeruli and

Global crescent proportion of glomeruli had important impacts on the predictability of the models. In future work, further prospective multicenter studies with multiple datasets are needed to evaluate the validity of these model and to reduce the influence of the imbalance in the target variables.

## Supporting information

**S1 Table. Features included demographic, clinical, laboratory data and treatment of the IgAN patients.**
(DOCX)

## Acknowledgments

The authors thank the Department of Nephrology of Guangdong Hospital of Traditional Chinese Medicine for their great support to this project. We acknowledge colleagues from Fane Data Technology (Tianjin) Corporation for providing technical support in machine learning model building.

## Author Contributions

**Conceptualization:** Chuan Zou.

**Data curation:** Yizhen Chen, Xiaodan Huang, Jundu Li, Yuansheng Hou, Miaoying Shen, Zaoqiang Lin, Ronglin Zhang.

**Formal analysis:** Xuefei Lin, Yongfang Liu, Songlin Hong.

**Funding acquisition:** Xusheng Liu, Chuan Zou.

**Investigation:** Xuefei Lin.

**Methodology:** Xuefei Lin, Songlin Hong.

**Project administration:** Xusheng Liu.

**Resources:** Haifeng Yang, Chuan Zou.

**Supervision:** Xusheng Liu.

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
