## [Decision Letter · Decision Letter 0]

30 Dec 2021

PONE-D-21-34774Prediction of prognosis in immunoglobulin a nephropathy patients with focal crescent by machine learningPLOS ONE

Dear Dr. Zou,

Thank you for submitting your manuscript to PLOS ONE. After careful consideration, we feel that it has merit but does not fully meet PLOS ONE’s publication criteria as it currently stands. Therefore, we invite you to submit a revised version of the manuscript that addresses the points raised during the review process.

Please, resolve carefully all the issues raised from Reviewer 1 including the rationale behind the selection of the variables in the model and the composite outcome. About this, it is not clear the importance of crescent in the model.

We look forward to receiving your revised manuscript.

Kind regards,

Fabio Sallustio, PhD

Academic Editor

PLOS ONE

Journal Requirements:

Reviewers' comments:

Reviewer's Responses to Questions

**Comments to the Author**

1. Is the manuscript technically sound, and do the data support the conclusions?

Reviewer #1: Partly

Reviewer #2: Yes

2. Has the statistical analysis been performed appropriately and rigorously? 

Reviewer #1: No

Reviewer #2: Yes

3. Have the authors made all data underlying the findings in their manuscript fully available?

Reviewer #1: Yes

Reviewer #2: Yes

4. Is the manuscript presented in an intelligible fashion and written in standard English?

Reviewer #1: No

Reviewer #2: Yes

5. Review Comments to the Author

Reviewer #1: In this paper the authors present a new method based on machine learning for the prediction of prognosis in IgA nephropathy patients with focal crescent.

Considering the growing interest in machine learning and its applications in medicine, the paper is of sure interest for the scientific community.

However, some issues need to be solved before the paper is accepted for publication:

1. The paper is difficult to read. English need to be improved, and in the paragraphs the same concept is often repeated several times. Introduction is too long, as well as conclusions, and too much room is dedicated to provide info on ML. The paragraph “feature selection and model construction” need to be expanded.

2. At page 6, line 137, age > 80 years seems an exclusion criterion. However, this is not reported in the flow-chart (Fig. 2).

3. Page 7: if crescent depends on immunosuppression, why drug duration and dose were not recorded?

4. The combined event is quite confusing. It includes several ways to measure CKD progression, including different percentages of reduction of eGFR. In some cases, the reduction is an absolute number, in other measured at 1 or 2 years. Please explain the rationale behind this choice.

5. The predictive performance of the model was evaluated only by ROC curve. However, other parameters need to be taken into account to evaluate the performance of an algorithm. Please implement this section. Furthermore, in order to understand if crescent really improves the model performance, AUC with and without this variable should be reported.

6. In table 2 the authors report the chosen variables using random forest. However, looking at the random forest, the selection criterion is not clear, as variables scored as more important have been discarded. Please clarify this issue.

Minor issues:

Pag.5, line 111: Transfer learning is not properly an algorithm of ML, but rather a method used to train the model.

Reviewer #2: Dear Author

Congratulations.

Your Manuscript is novel and proper.

My comment : Accept.

Kind Regards.

Dear Author

Congratulations.

Your Manuscript is novel and proper.

My comment : Accept.

Kind Regards.

6. PLOS authors have the option to publish the peer review history of their article (what does this mean?). If published, this will include your full peer review and any attached files.

Reviewer #1: No

Reviewer #2: No

---

## [Author Response · Author response to Decision Letter 0]

8 Feb 2022

Dear Editor Fabio Sallustio and Reviewers,

On behalf of my co-authors, we greatly appreciate the careful review and comments from both you and the reviewers. We believe that by implementing the suggested changes, we now have a stronger manuscript entitled “Prediction of prognosis in immunoglobulin a nephropathy patients with focal crescent by machine learning” for submission to PLOS ONE. We look forward to your positive response to the revised work submitted here.

We present here point-to-point responses for each of the comments in the attached document and have revised our manuscript accordingly. We do not change our statistics or results. And we hope the revised manuscript could be acceptable for you. Revised sections are identified with red text in the paper.

There are no conflicts of interest regarding this work. All authors have read the revised manuscript and approved its submission to PLOS ONE. Please do not hesitate to contact us if we can be of any further assistance.

Thank you and best regards. 

Sincerely yours,

Xusheng Liu, Chuan Zou.

Department of Nephrology, Guangdong Provincial Hospital of Chinese Medicine, Guangzhou, Guangdong, China;

Correspondence: Xusheng Liu, liuxu801 @ 126.com; Chuan Zou, doctorzc541888 @ 126.com

Response to the editor’s comments:

Dear Editor Fabio Sallustio,

Thank you very much for reviewing our manuscript. The main corrections in the manuscript text and our responses to your comments are as following:

Comment1: When submitting your revision, we need you to address these additional requirements. Please ensure that your manuscript meets PLOS ONE's style requirements, including those for file naming. The PLOS ONE style templates can be found at https://journals.plos.org/plosone/s/file?id=wjVg/PLOSOne_formatting_sample_main_body.pdf and https://journals.plos.org/plosone/s/file?id=ba62/PLOSOne_formatting_sample_title_authors_affiliations.pdf

Response1: Thanks for your kind suggestion. Our manuscript meets PLOS ONE's style requirements.

Comment2: Please note that PLOS ONE has specific guidelines on code sharing for submissions in which author-generated code underpins the findings in the manuscript. In these cases, all author-generated code must be made available without restrictions upon publication of the work. Please review our guidelines at https://journals.plos.org/plosone/s/materials-and-software-sharing#loc-sharing-code and ensure that your code is shared in a way that follows best practice and facilitates reproducibility and reuse.

Response2: Thanks for your kind suggestion. The Python code underlying reported findings were deposited in appropriate public data repository which is Figshare. The DOI is 10.6084/m9.figshare.19127399.

Comment3: In your Data Availability statement, you have not specified where the minimal data set underlying the results described in your manuscript can be found. PLOS defines a study's minimal data set as the underlying data used to reach the conclusions drawn in the manuscript and any additional data required to replicate the reported study findings in their entirety. All PLOS journals require that the minimal data set be made fully available. For more information about our data policy, please see http://journals.plos.org/plosone/s/data-availability.

Response3: Thanks for your kind suggestion. Our study’s data set underlying reported findings were deposited in appropriate public data repository which is Figshare. The DOI is 10.6084/m9.figshare.19127342.

Responds to the reviewer’s comments:

Reply to Reviewer #1

Dear Reviewer, 

Thank you for your positive comments and valuable suggestions to improve the quality of our manuscript.

Comments:

In this paper the authors present a new method based on machine learning for the prediction of prognosis in IgA nephropathy patients with focal crescent.

Considering the growing interest in machine learning and its applications in medicine, the paper is of sure interest for the scientific community.

However, some issues need to be solved before the paper is accepted for publication:

1. The paper is difficult to read. English need to be improved, and in the paragraphs the same concept is often repeated several times. Introduction is too long, as well as conclusions, and too much room is dedicated to provide info on ML. The paragraph “feature selection and model construction” need to be expanded.

2. At page 6, line 137, age > 80 years seems an exclusion criterion. However, this is not reported in the flow-chart (Fig. 2).

3. Page 7: if crescent depends on immunosuppression, why drug duration and dose were not recorded?

4. The combined event is quite confusing. It includes several ways to measure CKD progression, including different percentages of reduction of eGFR. In some cases, the reduction is an absolute number, in other measured at 1 or 2 years. Please explain the rationale behind this choice.

5. The predictive performance of the model was evaluated only by ROC curve. However, other parameters need to be taken into account to evaluate the performance of an algorithm. Please implement this section. Furthermore, in order to understand if crescent really improves the model performance, AUC with and without this variable should be reported.

6. In table 2 the authors report the chosen variables using random forest. However, looking at the random forest, the selection criterion is not clear, as variables scored as more important have been discarded. Please clarify this issue.

 Comment1: The paper is difficult to read. English need to be improved, and in the paragraphs the same concept is often repeated several times. Introduction is too long, as well as conclusions, and too much room is dedicated to provide info on ML. The paragraph “feature selection and model construction” need to be expanded.

Response1: Thanks for your kind suggestion. We are very sorry for our poorly written manuscript. We improve our article and we hope the revised manuscript could be acceptable for you. And here we did not list the changes but marked in red in the manuscript.

Comment2: At page 6, line 137, age > 80 years seems an exclusion criterion. However, this is not reported in the flow-chart (Fig. 2).

Response2: Thanks for your suggestion. The number of patients older than 80 was zero. As suggested by the reviewer, we have corrected the sentence into “Age > 18 years old.” 

page 6, line 137;

Comment3: Page 7: if crescent depends on immunosuppression, why drug duration and dose were not recorded?

Response3: Thank you for the detailed review. This is an especially important issue. According to 2021 KDIGO guidelines, there is insufficient evidence to support the use of the Oxford MEST-C score in determining whether immunosuppression should be commenced in IgAN[1]. However, in clinical practice, crescents are often associated with the use of immunosuppressants. Due to the limitation of the retrospective study design, the duration and dosage of IS therapy were not collected. Therefore, more prospective studies with a larger cohort are needed to support the present findings, retrospective analyses alone are not sufficient to determine the treatment choice for patients with IgAN, see the last paragraph of the discussion section. 

Comment4: The combined event is quite confusing. It includes several ways to measure CKD progression, including different percentages of reduction of eGFR. In some cases, the reduction is an absolute number, in other measured at 1 or 2 years. Please explain the rationale behind this choice.

Response4: Thank you for your careful observations. In this study, we defined clinical outcome: the combined event (Doubling of serum creatinine, 50% reduction in eGFR, 15% reduction in eGFR within 1 year, 30% reduction in eGFR within 2 years, ESRD or death). The primary clinical outcome was ESRD or death. The US FDA currently accepts 50% reduction in eGFR, assessed as doubling of serum creatinine level, as a surrogate end point for the development of kidney failure in clinical trials of kidney disease progression. In addition, based on a series of meta-analyses of cohorts and clinical trials and simulations of trial designs and analytic methods, 30% reduction in eGFR within 2 to 3 years may be an acceptable surrogate end point in some circumstances[2]. However, the population in this study is IgAN with focal crescent formation and without obvious chronic renal lesions, which is mainly acute lesions. As a result, we added 15% reduction in eGFR within 1 year in the combined event.

Comment5: The predictive performance of the model was evaluated only by ROC curve. However, other parameters need to be taken into account to evaluate the performance of an algorithm. Please implement this section. Furthermore, in order to understand if crescent really improves the model performance, AUC with and without this variable should be reported.

Response5: Thank you again for your valuable suggestions to improve the quality of our manuscript. In addition to ROC Curve, we reported on Precision Recall Curve and LIFT Curve, as well as detailed Precision Recall Curve metrics for each model, further more, supplemented by a description of the Calibration curve Figure 9 and Figure10 in the article. 

Fig 9. Calibration plots of the three candidate models for the prognosis of IgAN with 'Crescent proportion' and 'Global crescent proportion'.

As suggested by the reviewer, we have added three models (IgAN without 'Crescent proportion' and 'Global crescent proportion' Models)(Table4) .Although SVM of IgAN without crescent model has higher AUROC, SVM of IgAN with crescent model has higher AUPRC and F1 values. In clinical practice, particularly for our imbalanced dataset, the AUPRC and F1-score are more practical evaluation indicators for ML. Based on this requirement, we preferred the SVM of IgAN with crescent model rather than other models.

Table 3. Summary of the comparison of IgAN with 'Crescent proportion' and 'Global crescent proportion' model performance

Prediction model Precision

 Recall

 F1-score

 Accuracy

 AUROC AUPRC

Support Vector Machine 0.77 0.77 0.73 0.77 0.7957 0.765

Random Forest 0.69 0.70 0.61 0.70 0.6443 0.472

Naïve Bayes 0.74 0.74 0.69 0.74 0.7078 0.637

Table 4. Summary of the comparison of IgAN without 'Crescent proportion' and 'Global crescent proportion' model performance

Prediction model Precision

 Recall

 F1-score

 Accuracy

 AUROC AUPRC

Support Vector Machine 0.78 0.68 0.56 0.68 0.831 0.716

Random Forest 0.65 0.68 0.63 0.68 0.7041 0.567

Naïve Bayes 0.70 0.72 0.70 0.72 0.5959 0.567

Fig 6. Receiver operating characteristic(ROC) curves of the three candidate models for the prognosis of IgAN without 'Crescent proportion' and 'Global crescent proportion'.

Fig 7. Precision-recall curves of the three candidate models for the prognosis of IgAN without 'Crescent proportion' and 'Global crescent proportion'. 

Fig 10. Calibration plots of the three candidate models for the prognosis of IgAN without 'Crescent proportion' and 'Global crescent proportion'.

Comment6: In table 2 the authors report the chosen variables using random forest. However, looking at the random forest, the selection criterion is not clear, as variables scored as more important have been discarded. Please clarify this issue.

Response6: Thank you for your kind suggestion. A total of 10 important features were selected as important predictors for modeling on the basis of data-driven and medical selection, as shown in Table 2.

The feature selection method of our modeling considers the following principles to select the features that participate in the modeling: 

1.1 The top features found by the feature selection algorithm. 

1.2 The selected features cover different aspects as far as possible, such as patient pathological characteristics, clinical characteristics, epidemiological characteristics, and so on. 

1.3 The selected features are as independent as possible, that is, to minimize the strong correlation between multiple variables. 

1.4 Generally speaking, the amount of data of a mathematical model should be at least 10 times the number of independent variables of the model, on the other hand, the number of features involved in modeling should not be too much. The mathematical model established by too many variables can be poorly explained. 

1.5 Focus on the principles and practical experience of the medical profession.

Therefore, based on the above principles, we selected the top ten features including: baseline estimated GFR, serum creatine, serum triglycerides, proteinuria, MAP and Hematuria, C3, age, crescents proportion of glomeruli, Global crescent proportion of glomeruli.

Other comments: 

Minor issues:

Pag.5, line 111: Transfer learning is not properly an algorithm of ML, but rather a method used to train the model.

Response: Thank you for your careful observations, we have amended it.

Reply to Reviewer #2

Dear Reviewer, 

Thank you very much for your time involved in reviewing the manuscript and your very encouraging comments on the merits.

Comments:

Dear Author

Congratulations.

Your Manuscript is novel and proper.

My comment : Accept.

Kind Regards. 

References:

[1] 2021 KDIGO 2021 Clinical Practice Guideline for the Management of Glomerular Diseases KIDNEY INT 100 S1-276

[2] Levey A S, Inker L A, Matsushita K, Greene T, Willis K, Lewis E, de Zeeuw D, Cheung A K and Coresh J 2014 GFR Decline as an End Point for Clinical Trials in CKD: A Scientific Workshop Sponsored by the National Kidney Foundation and the US Food and Drug Administration AM J KIDNEY DIS 64 821-35

---

## [Decision Letter · Decision Letter 1]

21 Feb 2022

Prediction of prognosis in immunoglobulin a nephropathy patients with focal crescent by machine learning

PONE-D-21-34774R1

Dear Dr. Zou,

We’re pleased to inform you that your manuscript has been judged scientifically suitable for publication and will be formally accepted for publication once it meets all outstanding technical requirements.

Kind regards,

Fabio Sallustio, PhD

Academic Editor

PLOS ONE

Additional Editor Comments (optional):

Reviewers' comments:

Reviewer's Responses to Questions

**Comments to the Author**

1. If the authors have adequately addressed your comments raised in a previous round of review and you feel that this manuscript is now acceptable for publication, you may indicate that here to bypass the “Comments to the Author” section, enter your conflict of interest statement in the “Confidential to Editor” section, and submit your "Accept" recommendation.

Reviewer #1: All comments have been addressed

2. Is the manuscript technically sound, and do the data support the conclusions?

Reviewer #1: Yes

3. Has the statistical analysis been performed appropriately and rigorously? 

Reviewer #1: Yes

4. Have the authors made all data underlying the findings in their manuscript fully available?

Reviewer #1: No

5. Is the manuscript presented in an intelligible fashion and written in standard English?

Reviewer #1: Yes

6. Review Comments to the Author

Reviewer #1: (No Response)

7. PLOS authors have the option to publish the peer review history of their article (what does this mean?). If published, this will include your full peer review and any attached files.

Reviewer #1: No

---

## [Editor Report · Acceptance letter]

28 Feb 2022

PONE-D-21-34774R1 

Prediction of prognosis in immunoglobulin a nephropathy patients with focal crescent by machine learning 

Dear Dr. Zou:

I'm pleased to inform you that your manuscript has been deemed suitable for publication in PLOS ONE. Congratulations! Your manuscript is now with our production department. 

Kind regards, 

on behalf of

Dr. Fabio Sallustio 

Academic Editor

PLOS ONE